# Circulating microRNAs as potential biomarkers of early vascular damage in vitamin D deficiency, obese, and diabetic patients

**Adel B. Elmoselhi**[1,2]*, **Mohamed Seif Allah**[1,3☯], **Amal Bouzid**[2☯], **Zeinab Ibrahim**[2], **Thenmozhi Venkatachalam**[2], **Ruqaiyyah Siddiqui**[4], **Naveed Ahmed Khan**[1,2], **Rifat A. Hamoudi**[1,2]

**1** College of Medicine, University of Sharjah, Sharjah, UAE, **2** Sharjah Institute of Medical Research, College of Medicine, University of Sharjah, Sharjah, UAE, **3** Cardiology Department, University Hospital Sharjah, Sharjah, UAE, **4** College of Arts and Sciences, American University of Sharjah, University City, Sharjah, UAE

☯ These authors contributed equally to this work.
* amoselhi@sharjah.ac.ae

**Data Availability Statement:** The raw datasets presented in this study can be found in the online

## Abstract

Vitamin D3 deficiency, obesity, and diabetes mellitus (DM) have been shown to increase the risk of cardiovascular diseases (CVDs). However, the early detection of vascular damage in those patients is still difficult to ascertain. MicroRNAs (miRNAs) are recognized to play a critical role in initiation and pathogenesis of vascular dysfunction. Herein, we aimed to identify circulating miRNA biomarkers of vascular dysfunction as early predictors of CVDs. We have recruited 23 middle-aged Emiratis patients with the following criteria: A healthy control group with vitamin D $\geq$ 20ng, and BMI < 30 (*C1 group = 11 individuals*); A vitamin D deficiency (Vit D level $\leq$ 20 ng) and obese (BMI $\geq$ 30) group (*A1 group = 9 patients*); A vitamin D deficiency, obese, plus DM (*A2 group = 3 patients*). Arterial stiffness via pulse wave velocity (PWV) was measured and the whole transcriptome analysis with qPCR validation for miRNA in plasma samples were tested. PWV relative to age was significantly higher in A1 group 19.4 ± 4.7 m/s and A2 group 18.3 ± 1.3 m/s compared to controls 14.7 ± 2.1 m/s (p < 0.05). Similar patterns were also observed in the Augmentation pressure (AP) and AIx%. Whole RNA-Sequencing revealed miR-182-5p; miR-199a-5p; miR-193a-5p; and miR-155-5p were differentially over-expressed (logFC > 1.5) in high-risk patients for CVDs vs healthy controls. Collectively, our result indicates that four specific circulating miRNA signature, may be utilized as non-invasive, diagnostic and prognostic biomarkers for early vascular damage in patients suffering from vitamin D deficiency, obesity and DM.

## Introduction

Cardiovascular diseases (CVDs) are the leading cause of mortality and morbidity, as well as an increasing financial burden almost all around the globe. Deaths attributed to CVDs were approximately estimated to be 19 million in 2020 globally, an increase of 18.7% from 2010. In

repository Figshare under accession number: 0774806.

**Funding:** This study was supported by Sheikh Hamdan Bin Rashid Al Maktoum Award for Medical Sciences, Ref: MRG/38 (ABE received the award). The funder had no role in study design, data collection and analysis, decision to publish, or preparation of the manuscript.

**Competing interests:** The authors have declared that no competing interests exist.

the United States, the estimated direct costs of CVDs have doubled to $226.2 billion in 2017–2018 [1].

Vitamin D3 deficiency has been shown in several studies to increase the risk of CVDs [2]. Moreover, obesity and overweight are considered as high-risk factors for CVDs [3]. Of note, in the United Arab Emirates (UAE) more than 30% of the adult population are suffering from obesity compared to 13% globally [4]. Furthermore, a cross-sectional study of obese and diabetic revealed a vast prevalence of 83% of vitamin D deficiency, in the UAE population [5]. Another study showed that only 2.5% of indoor employees have sufficient vitamin D levels [6]. Furthermore, vitamin D deficiency is well-known to be associated with various CVDs such as hypertension, myocardial infarction, stroke, congestive heart failure, peripheral vascular disease, and atherosclerosis [7].

Vitamin D3 is a steroid fat-soluble vitamin and is the active form of vitamin D mainly produced metabolically in the kidney as a result of skin exposure to sufficient ultraviolet B radiation, and a smaller amount is taken up via nutrition and absorbed from the gastrointestinal tract. Vitamin D3 has pleiotropic functions, including an anti-inflammatory effect that promotes the differentiation of monocytes to macrophages, lymphocytes and dendritic cells. Because these cells play an important role as the first line of defense in the immune system, vitamin D is important in infection control [8]. Vitamin D further enhances the immune system to combat inflammation via the differentiation of active CD4+ T cells and an increase in the inhibitory function of T cells [9]. Activation of local inflammatory cytokines has also been reported and is speculated to be due to a possible molecular mechanism that links vitamin D and endothelial dysfunction, atherogenesis development in coronary arteries as well as other CVDs [10]. Previously, vitamin D was shown to suppress the kappa-light-chain-enhancer nuclear factor of the activated B-cell (NF-κB) pathway. Activation of NF-κB occurs by targeting karyopherin subunit alpha 4 (KPNA4) which mitigates the progression of CVDs. Therefore, vitamin D deficiency enhances the regulation of KPNA4 and consequently increases the activation of NF-κB [11]. NF-κB binds to various κB elements in the nucleus and acts as a transcription factor that stimulates the transcription of inflammatory cytokines such as interleukin IL-6, IL-8 and tumor necrosis factor TNF-α [12].

Furthermore, vitamin D was shown to inhibit calcification in blood vessels via affecting interleukin activities [13]. Therefore, it is logical to hypothesize that vitamin D may improve endothelial dysfunction as its' deficiency is associated with inflammation. Importantly, several studies in the last few years have been investigating this molecular aspect of endothelial dysfunction [14]. The protective role of Vitamin D in CVDs may also be attributed to the discovery of vitamin D receptors (VDR) in cardiomyocytes, endothelial cells and smooth muscle cells among other cells [15,16]. VDR is an intracellular receptor that binds to the active form of vitamin D, 1,25(OH)2D3. Once bound, it joins the retinoid X receptor and translocates to the nucleus where it binds to VDRE, the regulator site of the element promotor region of DNA, which may promote the synthesis of proteins associated with vitamin D [17]. The pleiotropic effects of vitamin D occur by activating VDR in vascular endothelial cells and cardiomyocytes and regulating the renin-angiotensin system and pancreatic cell activity [18]. Thus, patients with vitamin D deficiency were associated with several cardiovascular and metabolic disorders such as arterial stiffness, endothelial dysfunction, left ventricular hypertrophy, and diabetes mellites among others [19,20].

Nonetheless, the majority of randomized controlled clinical studies did not reveal the beneficial effects of vitamin D on endothelial dysfunction [21,22]. However, most of these studies utilised lower doses of vitamin D supplements, approximately 400–800 IU/day for 16–24 weeks duration. The Institute of Medicine (IOM) recommendation is 600IU/day as a dietary allowance for ages < 70 years and 800IU/day for ages > 70. The basis of these

recommendations is mainly related to bone research and healthy conditions, but the optimum dose of vitamin D for other functions is not yet clear, although several researchers have investigated different doses and time frames [23,24]. An earlier study was conducted on nineteen obese adolescents aged 13–18, and vitamin D3 supplementation lasted for 12 weeks and showed increasing 25(OH)D levels with no effect on endothelial function [25]. Another study indicated that vitamin D3 supplementation improves arterial stiffness in a dose-response manner of up to ~4000 IU/day for 16 weeks in overweight African-Americans with vitamin D deficiency [26]. Further studies are still warranted to clarify the optimal dosage and duration of vitamin D supplementation to obtain benefits for vascular dysfunction and CVDs prevention.

Epigenetic mechanisms have been recently shown to regulate aging, genetic and lifestyle factors in determining the risk of vascular diseases [27]. Changes in the gene expression by epigenetic mechanisms include DNA methylation, post-translational histone modifications, and non-coding RNAs (ncRNAs) [28]. Increasing evidence in the last two decades points out the critical role of non-coding RNAs such as microRNA (miRNAs) and lncRNAs in various physiological and pathological conditions [29]. miRNAs are small RNA molecules with a powerful regulatory function for several biological processes since it regulates gene expression at the post-transcriptional level and affects protein translation [30]. It has been shown that miRNAs are present in the blood, plasma, erythrocytes and platelets [31]. These circulating endogenous miRNAs are very stable even under harsh conditions (e.g. extreme temperature) probably due to some protective mechanisms that prevent their degradation [32]. Therefore, circulatory miRNAs are now recognized to have a great potential to be used as biomarkers to predict and assess vascular diseases as well as possible novel therapeutic targets.

Since endothelial dysfunction is considered one of the early signs of CVDs, detection of endothelial dysfunction will have a significant positive impact on preventing CVDs, especially in high-risk individuals that suffer from vitamin D deficiency, obesity and diabetes mellitus (DM). Thus, there is a tremendous need to investigate and establish a prevention approach for early detection of cardiovascular complications, especially in high-risk individuals. Our study examined vascular dysfunction and determined its relevance in circulatory miRNAs and explored the whole transcriptome in adult Emiratis suffering from vitamin D deficiency and obesity, as well as patients with vitamin D deficiency, obesity and diabetes mellitus (DM) compared to a control group of participants with normal vitamin D levels and non-obese.

## Methods

### Ethics statement

This study was reviewed and approved by the University of Sharjah Research Ethics Committee with Reference number REC 16–11–12, as well as by the ethics committee of University Hospital Sharjah (UHS). The procedures used in this study fully adhered to the tenets of the Declaration of Helsinki. The patients/participants provided their written informed consent to participate in this study.

### Study population

All participants were recruited at University Hospital Sharjah (UHS), between December, 2018 and March, 2020, from the Cardiology and Family medicine clinics. Participants were all Emirati nationals, and the inclusion and exclusion criteria were determined to achieve the main objectives as described in Fig 1. Briefly, we targeted middle-aged Emirati nationals with and without vitamin D (Vit D) deficiency with no previously diagnosed cardiovascular disorders or any other major debilitating diseases, to mainly detect the early changes in their vasculature. In particular, the inclusion criteria consisted of males > 45-year-old and

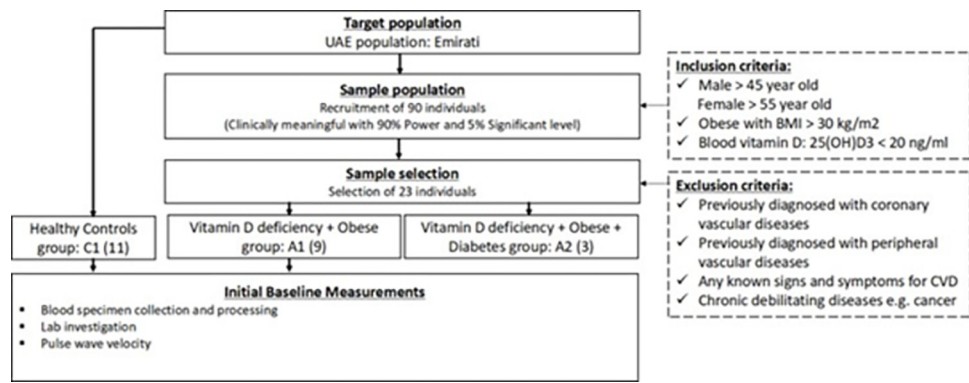

**Fig 1. Flow diagram of sample selection, inclusion and exclusion criteria, and baseline measurements.**

females > 55-year-old (to avoid the changes in hormonal influence in pre-menopausal women); obese with body index mass (BMI) of more than 30 kg/m$^2$; and blood Vit D, 25(OH) D3 status below 20 ng/ml. While the exclusion criteria comprised previously diagnosed with coronary vascular diseases; previously diagnosed with peripheral vascular diseases; any known signs and symptoms for CVDs; chronic debilitating diseases e.g. cancer. Out of the initial target of 90 participants, we managed to recruit 23, and the valid numbers included were: 11 controls (Vit D level > 20 ng, BMI < 30), 9 Vit D deficiency and obese (Vit D level ≤ 20 ng, BMI ≥ 30), and 3 Vit D deficiency, obese and diabetic (Vit D level ≤ 20 ng, BMI ≥ 30) as shown in Table 1. The low number of recruitments, which is considered as a limitation of the study, was due to the strict criteria of inclusion and exclusion, and the fact that the recruitment was conducted in only one university hospital that eventually was crippled by COVID-19 restrictions.

## Clinical investigations

Medical history, physical examination and routine laboratory investigation were obtained for all participants plus pulse wave velocity (PWV) for assessing vascular stiffness. PWV measurement has been described in detail elsewhere [33]. Briefly, PWV measurement was performed using SphygmoCor (version 7.0, Atcor Medical, Sydney, Australia) for individuals in the supine position after a minimum of 10 min rest in a quiet temperature-controlled room. The PWV of the 'aortic' segment (aortic PWV) was recorded between the femoral and carotid arteries. The distance from the carotid recording site to the suprasternal notch was subtracted from the distance between the femoral recording site to the suprasternal notch. The quality demands of PWV were followed as suggested by the manufacturer, showing that a quality index ≥ 80% is accepted. The PWV relative to age was calculated as follows: PWV (m/s) x 100 / Age (year old).

## Plasma samples collection

In the EDTA vacutainer tube, 10 ml of whole blood was collected from each participant. 2 ml of whole blood was transferred to a separate labeled tube and immediately placed in a -80˚C freezer. Next, we centrifuged the rest of the whole blood at 4200 RPM for 10 minutes. 4 ml of the plasma was transferred to a tube and immediately placed in a -80˚C freezer.

## Extraction of mRNA and miRNA from plasma samples

First, mRNA was isolated according to the TRIzol protocol (Invitrogen, USA), as described elsewhere [34]. In brief, in a 1.5ml microfuge tube, 1000 μl Trizol was added to 200 μl of whole

**Table 1. Clinical Characteristics of three groups: Control (C1 group) who have normal vitamin D levels and non-obese, patients who suffer from vitamin D deficiency and obesity (A1 group), patients who suffer from vitamin D deficiency, obesity, and diabetes mellitus (A2 group).**

| Parameters | Control (C1 group) | Patients (A1 group) | p-value (C1 vs A1) | Patients (A2 group) | p-value (C1 vs A2) | p-value (A1 vs A2) |
|---|---|---|---|---|---|---|
| **Age** | 58.4 ± 7.5 | 53 ± 11.7 | 0.1934 | 63 ± 7 | 0.1889 | 0.1889 |
| **Gender, M/F** | 5 (45.5%)/ 6 (54.5%) | 6 (66.6%)/ 3 (33%) | - | 0/3 (100%) | - | - |
| **Plasma Vit. D level** | 37.4 ± 12.6 | 12.9 ± 4.3 | <0.0001*** | 11.7 ± 1.2 | 0.005** | 0.4657 |
| **BMI** | 26.4 ± 2.6 | 33.6 ± 3.1 | <0.0001*** | 34.9 ± 0.9 | 0.0002*** | 0.6382 |
| **Brachial systolic pressure, mmHg** | 127.5 ± 13.5 | 124.5 ± 8.8 | 0.5563 | 129.3 ± 14.3 | 0.8359 | 0.4745 |
| **Brachial diastolic pressure, mmHg** | 78.5 ± 8.1 | 81 ± 7.3 | 0.4636 | 70.3 ± 10.3 | 0.1621 | 0.0603 |
| **Aortic systolic pressure, mmHg** | 119 ± 15.2 | 115.1 ± 10.9 | 0.5107 | 118.7 ± 9 | 0.9721 | 0.6182 |
| **Aortic diastolic pressure, mmHg** | 79.6 ± 8.1 | 81.9 ± 7.4 | 0.5137 | 71.3 ± 9.3 | 0.1506 | 0.0641 |
| **Augmentation pressure (AP), mmHg** | 10 ± 3.2 | 12 ± 5.01 | 0.3930 | 14 ± 4.4 | 0.1363 | 0.5704 |
| **AIx%** | 24.1 ± 7 | 27 ± 9.5 | 0.5224 | 28.3 ± 0.6 | 0.3407 | 0.8193 |
| **PWV (m/s)** | 7.7 ± 1 | 10.1 ± 3.1 | 0.0790 | 11.5 ± 0.69 | 0.0206* | 0.4739 |
| **PWV relative to age** | 14 ± 2.1 | 19.4 ± 4.7 | 0.0173* | 18.3 ± 1.3 | 0.0284* | 0.7107 |

BMI: Body mass index, AP: Augmentation pressure, AIx%: AP/pulse wave pressure, PWV: Pulse Wave Velocity

*: Significantly different (p ≤ 0.05)

**: Significantly different (p ≤ 0.005)

***: Significantly different (p ≤ 0.0005).

blood. Then, the whole was incubated for 20 minutes on ice and 200 μl of chloroform was added. After 10 mins centrifugation, the upper aqueous solution was transferred to a new microfuge tube. To pellet the RNA, 500 μl of 100% isopropanol was added, followed by 10 minutes of incubation at room temperature and 10 minutes of centrifugation. Following discarding the supernatant and washing steps, the pellet was air-dried and re-suspended in 30 μl RNase-free water. Second, miRNA was isolated using miRNeasy Serum/Plasma Kit (Qiagen, Valencia, CA) according to the manufacturer's protocol.

## Whole RNA library preparation and sequencing

Whole transcriptome sequencing was performed for all 23 patients using RNA-Sequencing with Ion AmpliSeq Whole Transcriptome human gene expression kit (Thermo Fisher Scientific, Massachusetts, USA). Briefly, the purified RNAs were evaluated and quantified. Spectrophotometry was performed on the samples and the ratio of absorbance at 260 nm and 280 nm (A260/A280) was used to assess the purity of RNA. A ratio of around 2.0 indicated pure RNA without protein or organic contamination. cDNA library was constructed using Turbo DNase treated RNA using SuperScipt VILO cDNA synthesis kit (Invitrogen, Life Technologies, CA, USA). Samples were then tagged with unique Ion express barcodes and purified using AMPure XP Reagent (Beckman Coulter, USA). Barcoded libraries were assessed using the Ion Taqman

library quantitation kit (Thermo Fisher Scientific) and then pooled equally. The pooled libraries were amplified using emulsion PCR on Ion One Touch2 instruments (OT2) and enriched using Ion One Touch ES as per the manufacturer's instructions. The constructed cDNA libraries were sequenced using an Ion 540 Chip on an Ion S5 XL Semiconductor sequencer (Life Technologies).

## Transcriptomic data processing

The generated RNA-Seq data were processed using an in-house pipeline and analyzed using DESeq2 R/Bioconductor package to identify the differentially expressed genes in each comparison of patients with Vit D deficiency and obesity (Vit D level $\leq$ 20 ng, BMI $\geq$ 30) (A1 group) and Vit D deficiency, obese and diabetic (Vit D level $\leq$ 20 ng, BMI $\geq$ 30) (A2 group) against control healthy individuals with Vit D level > 20 ng and BMI < 30 (C1 group). Differentially overexpressed genes were selected with Fold change (FC) log2(FC) $\geq$ 1.5.

## Target miRNA identification

In order to identify circulating miRNAs that could serve as biomarkers of vascular dysfunction in high-risk patients with vitamin D deficiency, obesity and MD, a comprehensive bioinformatics analysis was performed using the transcriptomic data. The whole miRNA selection process was described in the flowchart in Fig 2. The up-expressed genes in patients compared to controls were reverse mapped to identify the target miRNAs. First, we were interested in the genes that are uniquely over-expressed in each of the subgroups and with (FC) log2(FC) $\geq$ 1.5. Genes with low read counts were filtered out; only genes with more than 30 normalized read counts were considered for further analysis. The predicted miRNA targets were identified using TargetScan (https://www.targetscan.org/vert_80/) and miRDB (http://mirdb.org/) databases. The miRNAs were then selected based on; the most prevalent transcript, a high

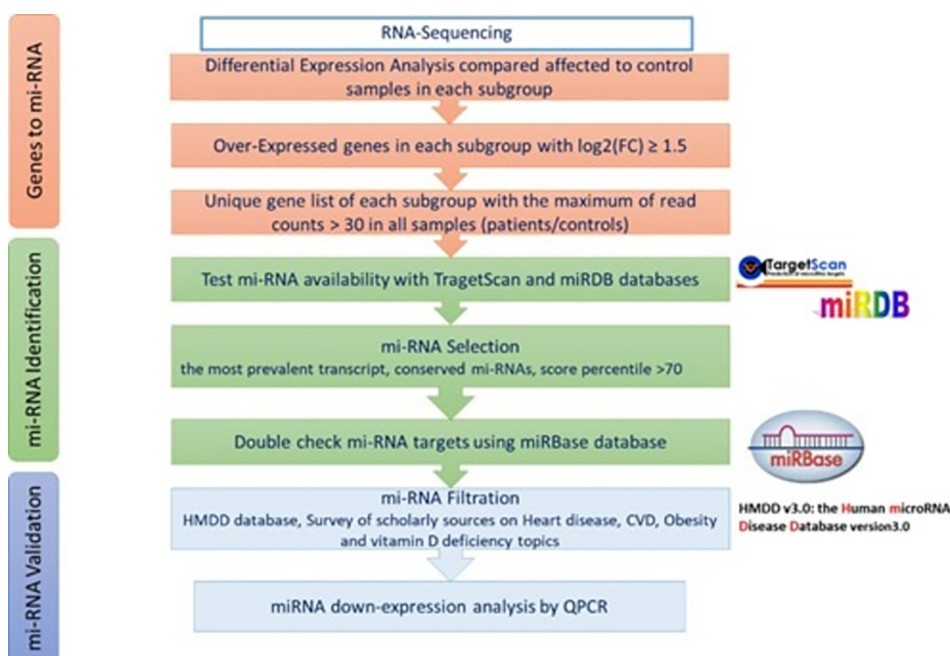

**Fig 2. Flow chart of gene targets pipeline and miRNA selection process with color coding of genes to miRNA (orange), miRNA identification (green), and miRNA validation (purple).**

conservation prediction and a context score percentile of > 70. Poorly conserved miRNAs were not considered. All predicted miRNAs were further cross-checked using another micro-RNA database: miRbase (https://www.mirbase.org/). Then, the Human microRNA Disease Database (HMDD) (http://www.cuilab.cn/hmdd) was searched for all predicted miRNAs. Only miRNAs that showed experiment-supported evidence and disease associations with heart disease, CVDs, obesity, vitamin D deficiency and MD were selected as potential biomarkers.

## Validation of plasma miRNA using Quantitative real-time PCR

For plasma miRNA validation, cDNA was synthesized using microScript microRNA cDNA Synthesis Kit (Norgen, Canada) according to the manufacturer's protocol. The cDNA concentration and quality were measured by NanoDrop (Thermo Fisher). qRT-PCR was performed in a Qiagen Rotor-Gene qPCR system using miScript SYBR Green PCR Kit (Qiagen). The Ct-value of the miRNA of interest was normalized against the expression of the housekeeping miRNA (miR-19b-3p) from each sample. The relative gene expression was calculated using the $2^{-\Delta\Delta Ct}$ method [35]. The primer sequences are described in Table 2.

## Statistical analysis

Statistical analysis was performed using GraphPad Prism v.5.0 (GraphPad Software, Inc. CA, USA). The clinical parameters were presented in mean ± standard deviation (SD). The unpaired t-test was used to compare statistical significance between two groups (Group A1 vs. group C1; group A2 vs C1, and group A1 vs. A2). Mann–Whitney U-tests were used to calculate the significance of differentiation of any two patient groups by various miRNAs. miRNA expression was presented as mean ± standard error. Pearson correlation testing was carried out to determine correlations between miRNA expression and various clinical characteristics in all patients of group A1 and group A2. A two-tailed p-value (p) $< 0.05$ was considered statistically significant for all statistical tests.

## Results

### Clinical characteristics of study participants

The clinical parameters of vitamin D deficiency and obese (A1 group) and DM (A2 group) compared to control individuals with vitamin D level > 20 ng and BMI < 30 (C1 group) are presented as means ± SD in Table 1. The age of the A1 group was 53 ± 11.7, while the C1 group was 58.4 ± 7.5. Blood Vit D levels showed a significant difference between A1 vs. C groups (p $< 0.0001$) in A1 group was 12.9 ± 4.3 ng, while in C1 group was 37.4 ± 12.6 ng. BMI was significantly different between the two groups (p $< 0.0001$); in A1 group the BMI was 33.6 ± 3.1, while in C1 group it was 26.4 ± 2.6. A significant increase in vascular stiffness measured in PWV relative to the participant's age was observed in the A1 group compared to the

**Table 2. Primer's sequences for miRNAs validation by quantitative PCR.**

| miRNA name | Forward primer | Reverse Primer |
|---|---|---|
| hsa-miR-193a-5p | TCTTTGCGGGCGAGATG | GAACATGTCTGCGTATCTC |
| hsa-miR-182-5p | GGCAATGGTAGAACTCAC | GAACATGTCTGCGTATCTC |
| hsa-miR-200c-3p | GTCTTACCCAGCAGTGT | GAACATGTCTGCGTATCTC |
| hsa-miR-19b-3p | TGCAGGTTTGCATCCAG | GAACATGTCTGCGTATCTC |
| hsa-miR-199a-5p | CCAGTGTTCAGACTACC | GAACATGTCTGCGTATCTC |
| hsa-miR-155-5p | TGCTAATCGTGATAGGGG | GAACATGTCTGCGTATCTC |

**Table 3. Potential miRNAs biomarkers for early vascular damage in patients suffering from vitamin D deficiency, obesity and DM.**

| Comparison groups | mi-RNA | QPCR FC | Target gene | RNA-Seq FC |
|---|---|---|---|---|
| A1 vs C1 | miR-182-5p | -1.524 | CFL1 | 1.57 |
| A1 vs C1 | miR-200c-3p | -15.2 | KIAA1432 | 1.67 |
| A1 vs C1 | miR-199a-5p | -1.922 | ZNF415 | 2.1 |
| A2 VS C1 | miR-193a-5p | -70.763 | MTRNR2L8 | 1.77 |
| A1 vs A2 | miR-155-5p | -87.674 | C9orf78 | 1.54 |

A1 vs C1: Affected individuals with Vitamin D deficiency and Obesity (A1) compared to controls (C1). A2 vs C1: Affected individuals with Vitamin D deficiency, Obesity and Diabetes (A2) compared to controls (C1). A1 vs A2: Affected individuals with Vitamin D deficiency and Obesity (A1) compared to Affected individuals with Vitamin D deficiency, Obesity and Diabetes (A2).

control group with 19.4 ± 4.7 vs. 14.7 ± 2.1 m/s (p < 0.05). A similar trend was also noticed in ALX% (AP/pulse wave pressure) with 27.3 ± 9.5 vs. 24.5 ± 7 respectively. Augmentation pressure (AP), aortic/brachial systolic pressures and aortic/brachial diastolic pressures were also measured (Table 1). Furthermore, A2 group has shown a more significant increase in PWV relative to age compared to C1 group 18.3 ± 1.3 vs. 14.7± 2.1 m/s (P < 0.05), respectively. A similar trend was also noticed in ALX% between A2 group and C1 group 27 ± 9.5 vs. 24.1 ± 7 respectively, as well as in brachial systolic pressure and augmentation pressure.

## Identification of miRNAs in the plasma of high-risk patients for vascular damage

Based on our transcriptomic data, bioinformatics analysis was conducted comparing the three groups (C1, A1, and A2) to identify the potential miRNAs that are particularly associated with cardiovascular and metabolic disorders. The top miRNAs observed were miR-182-5p; miR-200c-3p; miR-199a-5p; miR-193a-5p; miR-155-5p. Between groups A1 and C1, the target genes of three miRNA -182-5p, miRNA-200-3p, and miRNA-199a-5p were CFL1, KIAA1432, and ZNF415, respectively as shown in Table 3. The target gene CFL1 was a 2.5 log2 fold increase in the A1 group compared to C1. This gene is described as Cofilin 1 in Gene Ontology (GO), in which its biological process is associated with actin filament fragmentation and depolymerization as well as regulation by the host viral process as shown in Table 4. Target gene KIAA1432 was a 1.66 log2 fold increase in the A1 group. It is described as RICI homolog RAB6A GEF complex partner and associated with the regulation of extracellular matrix constituent secretion and extracellular matrix organization. The target gene ZNF415 was a 2 log2 fold increase in the A1 group compared to the C1 group. The gene is described as Zinc finger protein 415 and is associated with regulation of transcription by RNA polymerase II and regulation of transcription, DNA template as shown in Table 4. For miRNA 193-5p, the target gene MTRNR2L8 was shown to increase 1.76 log2 fold in group A2 compared to the C1 group. This gene was described as MT-RNR2-like 8 and linked to the regulation of the execution phase of apoptosis and regulation of signaling receptor activity as shown in Table 4. In miRNA-155-5p, the target gene C9orf78 was shown to increase 1.53 log2 fold in group A1 compared to group A2. This gene was described as chromosome 9 open reading frame 78 and linked to the regulation of mRNA splicing via spliceosome, mRNA processing, and RNA splicing (Table 4).

## Validation of the down expression profiling of the top miRNAs

Validation of the top selected miRNAs was confirmed using qRT- PCR with the reference mir-19 as a normalization control since it has been reported to be stably expressed in different

**Table 4. miRNA-Target gene annotation.**

| mi-RNA | Target gene | log2 Fold Change | Gene Description | Biological Process (GO) |
|---|---|---|---|---|
| miR-182-5p | CFL1 | 2.50 | Cofilin 1 | GO:0030043 actin filament fragmentation;<br>GO:0030042 actin filament depolymerization;<br>GO:0044794 positive regulation by host of viral process |
| miR-200c-3p | KIAA1432 | 1.66 | RIC1 homolog, RAB6A GEF complex partner 1 | GO:0003330 regulation of extracellular matrix constituent secretion;<br>GO:0070278 extracellular matrix constituent secretion;<br>GO:1903053 regulation of extracellular matrix organization |
| miR-199a-5p | ZNF415 | 2.00 | Zinc finger protein 415 | GO:0006357 regulation of transcription by RNA polymerase II;<br>GO:0006366 transcription by RNA polymerase II;<br>GO:0006355 regulation of transcription, DNA-templated |
| miR-193a-5p | MTRNR2L8 | 1.76 | MT-RNR2 like 8 | GO:1900118 negative regulation of execution phase of apoptosis;<br>GO:1900117 regulation of execution phase of apoptosis;<br>GO:2000272 negative regulation of signaling receptor activity |
| miR-155-5p | C9orf78 | 1.53 | chromosome 9 open reading frame 78 | GO:0048024 regulation of mRNA splicing, via spliceosome;<br>GO:0050684 regulation of mRNA processing;<br>GO:0043484 regulation of RNA splicing |

log2 Fold Change: Patients compared to controls.

tissues and cell types, (Fig 3) [36,37]. We standardized the quantitative cycle (Cq) values based on the standardized Cq values of each miRNA. The lower the Cq value the higher the expression of the indicated miRNA in the plasma of the subject.

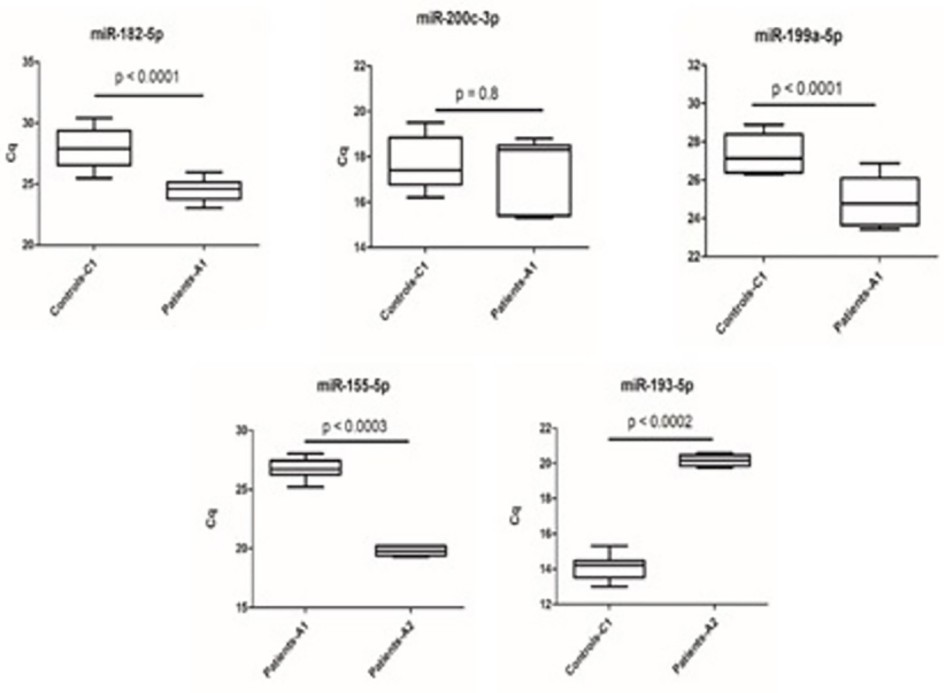

**Fig 3.** qRT-PCR validation for individual miRNA: Expression levels of miR-182-5p, miR-200c-3p, miR-199a-5p (A-C) are between Vit D deficiency and obese patients (A1 group) compared to control individuals (C1 group); miR-193a-5p expression level (D) is between Vit D deficiency, obese, and diabetes mellitus (A2 group) and control individuals (C1 group); and miR155-5p expression level (E) is between A1 group and A2 group.

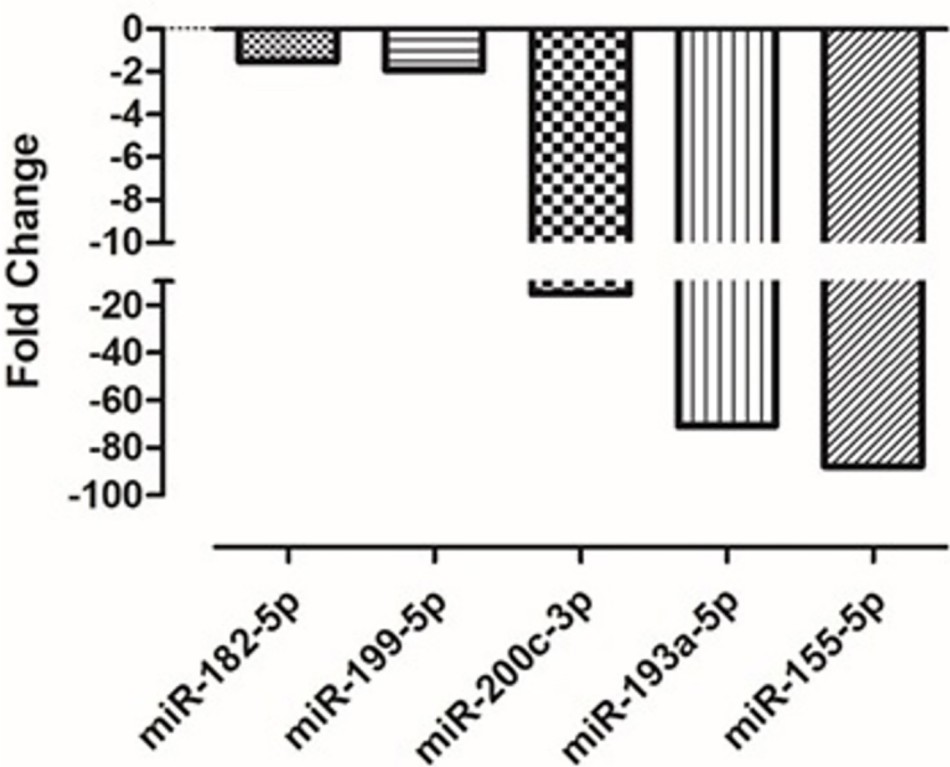

**Fig 4. miRNA down-expression validation by qPCR in affected patients compared to healthy control individuals.**
The gene expression comparison was between A1 and C1 groups for miR-182-5p, miR-200c-3p, miR-199a-5p; between A2 and C1 groups for miR-193p; and between A1 and A2 groups for miR155-5p expression.

## MiRNA 182-5p expression

Validation by qPCR has confirmed the down expression of miRNA 182-5p by showing a lower Cq in the A1 group compared to the C1 group. The expression of miRNA 182-5p was significantly (p < 0.0001) down expressed in vitamin D deficiency and obese patients (A1 group) by 1.524 folds compared to the control patients (C1 group) with normal vitamin D level and non-obese (Fig 4).

## MiRNA 200c-3p expression

The validation by qPCR showed lower expression in the A1 group compared to the C1 group as indicated in a higher quantitation cycle in the A1 group relative to the C1 group. The expression of miRNA 200c-3p was down expressed, but not statistically significant (p > 0.5), in vitamin D deficiency and obese patients (A1 group) by 15.2 folds compared to control patients (C1 group) with normal vitamin D level and non-obese (Fig 4).

## MiRNA 199a-5p expression

qPCR validation confirmed the down expression of miRNA 199a-5p in the A1 group compared to the C1 group by a significantly lower Cq in the A1 group compared to the C1 group as shown in Fig 4. The expression of miRNA 199a-5p was significantly lower (p < 0.0001) in vitamin D deficiency and obese patients (A1 group) by 1.992 folds compared to the control patients (C1 group) with normal vitamin D levels and non-obese.

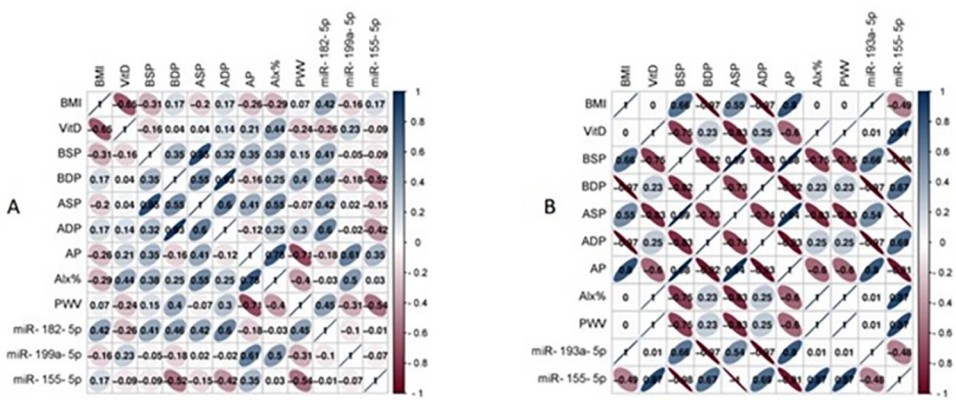

**Fig 5. Correlation matrix among circulating miRNA expression and various clinical parameters in (A) patients with vitamin D deficiency and obesity and (B) patients with vitamin D deficiency, obesity, and diabetes mellitus.** Analysis was performed by Pearson's Correlation, p ≤ 0.05. BMI: Body Mass Index; VitD: Vitamin D level (ng/ml); BSP: Brachial systolic pressure (mmHg), BDP: Brachial diastolic pressure (mmHg), ASP: Aortic systolic pressure (mmHg), ADP: Aortic diastolic pressure (mmHg), AP: Augmentation pressure (mmHg), Alx%: AP/pulse wave pressure; PWV: Pulse wave pressure (m/s).

## MiRNA 155-5p expression

The qPCR validation confirms the down expression of miRNA 155-5p in the A1 group compared to the A2 group by a lower quantitation cycle in the A2 group compared to the A1 group as shown in Fig 4. The miRNA 155-5p was very significantly down expressed (p < 0.0003) in vitamin D deficiency and obese patients (A1 group) by 87.674 folds compared to the patients suffering from vitamin D deficiency, obesity plus diabetes mellitus (A2 group).

## MiRNA 193-5p expression

The qPCR validation confirmed lower expression of miRNA 193-5p in the A2 group compared to the C1 group by a higher quantitation cycle in the A2 group compared to the C1 group (Fig 4). The expression of miRNA 193-5p was significantly lower (p < 0.0002) in vitamin D deficiency, obese patients plus diabetes mellitus (A2 group) by 70.763 folds compared to the control patients (C1 group).

### Circulating miRNA biomarkers correlate to vascular damage in high-risk patients

The matrix correlation between measured clinical parameters such as BMI, brachial and aortic systololic and diastolic blood pressure, Augmentation pressure (AP), Alx%, and PWV with circulating miRNAs, in particular miR-182-5p, miR-199a-5p, miR-155-5p, miR-193a-5p, and miR-155-5p as shown in Fig 5. Using Pearson's correlation in data analysis; of particular interest the higher correction between PWV and Alx% were miR-182-5p, miR-199a-5p, and miR-155-5p. Also, the higher correlation of miR-182-5p, miR-193a-5p and miR-155-5-5p with brachial and aortic blood pressure.

## Discussion

Vascular dysfunction as an early sign of CVDs has been associated with vitamin D deficiency. However, its accurate and early diagnosis is difficult to achieve. Measuring circulatory miRNAs as a non-invasive diagnostic and prognostic marker for early vascular dysfunction may be

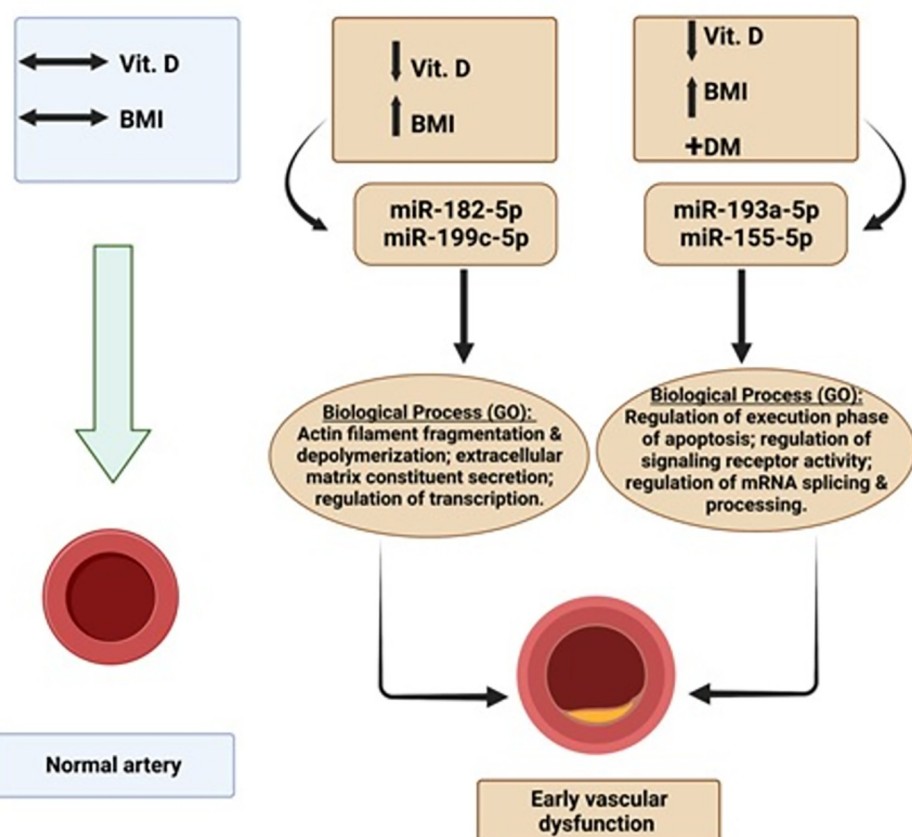

**Fig 6. Schematic summary for the novel circulatory miRNAs in patients suffering from vitamin D (Vit.D), obesity (high BMI), and diabetes mellitus (DM) which lead to early vascular dysfunction via expression of specific target genes with various biological processes via Gene Ontology (GO).**

a promising approach. Here, we report an increase in vascular stiffness in middle-aged obese Emirati patients suffering from vitamin D deficiency compared to control individuals with a normal level of vitamin D and non-obese, as well as to patients suffering from vitamin D deficiency, obesity, plus diabetes mellitus (DM). Furthermore, we have identified two miRNAs (miR-182-5p and miR-199a-5p) as promising biomarkers for vascular dysfunction in vitamin D deficiency and obese patients. Moreover, two other miRNAs (miR-193a-5p; miR-155-5p) have resulted in vascular dysfunction in patients suffering from vitamin D deficiency, obesity, and DM (Fig 6).

Arterial stiffness has been reported to be an independent predictor of CVDs and mortality. Although arterial stiffness increases with aging, its progression increases further in patients suffering from vitamin D deficiency [38]. We have not observed statistically significant changes among our groups in both brachial and aortic pressures, most likely because of the small sample size. Nonetheless, the aortic systolic pressure was consistently lower than brachial systolic pressure as previously reported, which is more indicative of the actual afterload the heart has to encounter [38].

miR-182-5p has been reported in several studies to be associated with heart failure as prognostic biomarkers. In particular, miR-182-5p was found with miR-200a and miR-568 to be inversely correlated with Left ventricular mass index (LVMI) [39]. The target gene for miR-182-5p is CFL1 in Homo sapiens (Human) for Cofilin-1 protein (https://www.uniprot.org/

uniprot/P23528). The function of the CFL1 gene includes the regulation of cell morphology and cytoskeletal organization in epithelial cells, as well as chemokine receptor ACKR2 up-regulation from the endosomal compartment to the cell membrane [40,41].

Downregulation of miR-199a-5p was shown in myocardial tissue of patients following coronary artery bypass graft surgery. It was also linked with an elevated level of cardioprotective protein Sirtuin 1 (SIRT 1) and in major adverse cardiac and cerebrovascular events at 3-year follow-ups [42]. Furthermore, alteration of miR-199a-5p has been associated with fibrosis and hypertrophic growth in diabetic cardiomyopathy, as well as with obesity and heart failure. The target gene of miR-199a-5p is ZNF415 which is the coding gene of the Zinc Finger Protein. It was reported to be differently expressed in vascular tumor-derived endothelium [43].

Comparing patients suffering from vitamin D deficiency, obesity and type 2 diabetes mellitus to control patients, significantly lower expression of miR-193a-5p was observed in our study. miR-193a-5p was also shown to be upregulated in heart hypertrophy for both *in-vitro* and *in-vivo* mice models induced by Ang II [44]. More importantly, it was associated in several studies with obesity, diabetes mellitus, as well as normoglycemic and hyperglycemic-affected adipogenesis [45]. In particular, mir-193a-5p was shown to fine-tune the adverse events of the MAPK signaling pathway to fight against obesity [46]. The miR-193a-5p target gene is *MTRNR2L8*. Recently, it has been reported that DNA methylation of *MTRNR2L8* may play an important part in large-artery atherosclerotic stroke. Thus, it was suggested as a potential therapeutic target and diagnostic biomarker for stroke [47]. Further comparison between vitamin D and obese patients and vitamin D deficiency, obese and diabetic patients was a significantly higher expression of miR-155-5p in group A2 vs group A1. Changes in miR-155-5p were reported in several studies in both animal and human models related to diabetes mellitus and cardiovascular pathogeneses. A recent study has reported that miR-155-5p upregulation affects myocardial insulin resistance via mTOR signaling in chronic alcohol-drinking rats [48]. Additionally, miR-155 deletion in female mice increases adipogenic, insulin sensitivity, and energy uncoupling machinery, while limiting inflammation in white adipose tissue, which together could restrict high-fat diet-induced fat accumulation. These results identify miR-155 as a target candidate for improving obesity resistance [49]. Another study on humans has reported vitamin D-mediated attenuation of miR-155 in macrophages infected with dengue virus, which was implicated due to the cytokine response [50]. The target gene of miR-155-5p is C9orf78, which has been associated with several human cancer types and consider as a good prognostic maker for patient survival (https://www.proteinatlas.org/humanproteome/pathology).

On the whole, the new miRNAs we have identified here are consistent with previous studies in their association with cardiovascular and metabolic effects. The sample size was a limitation in our study because of the specific inclusion and exclusion criteria as detailed in the method section. However, we were able to overcome this limitation by using different bioinformatics tools to select and validate potential differentially expressed miRNAs. Further studies are warranted to confirm and refine the early detection of vascular dysfunction, especially in high-risk individuals with vitamin D deficiency, obesity and diabetes mellitus among the Emirati population. Such studies are essential in order to prevent and mitigate more serious consequences of cardiovascular and cerebrovascular diseases.

## Conclusion

Vascular stiffness is associated with vitamin D deficiency, obesity, and diabetes mellitus. Early detection of vascular dysfunction could prevent CVD, especially in high-risk individuals. The results of our study confirm the link between vitamin D deficiency, obesity, DM, and vascular dysfunction, as well as identifying novel mi-RNAs biomarkers for early diagnosis of vascular

damage, especially for high-risk individuals. Herein we reported novel circulatory miRNAs biomarkers that are likely to provide specific non-invasive early diagnostic and prognostic tools for assessing vascular health. These comprise miR-182-5p and miR-199a-5p for vitamin D deficiency and obese patients, while miR-193a-5p and miR-155-5p for vitamin D, obese and diabetic patients. Considering the tremendous benefits of these novel biomarkers to prevent vascular complications in vitamin D deficiency, obese and diabetic patients, further larger studies are warranted. Finally, further research is needed to determine the precise role of vitamin D in CVD, in particular within the Emirati population.

## Acknowledgments

We would like to thank all the participants enrolled in this study. Moreover, we would like to thank Ms. Juliet De Leon for her imaging technical assistant.

## Author Contributions

**Conceptualization:** Adel B. Elmoselhi, Mohamed Seif Allah.

**Data curation:** Mohamed Seif Allah, Amal Bouzid, Zeinab Ibrahim, Thenmozhi Venkatachalam, Rifat A. Hamoudi.

**Formal analysis:** Adel B. Elmoselhi, Amal Bouzid, Thenmozhi Venkatachalam, Rifat A. Hamoudi.

**Funding acquisition:** Adel B. Elmoselhi.

**Investigation:** Mohamed Seif Allah, Zeinab Ibrahim, Thenmozhi Venkatachalam.

**Methodology:** Zeinab Ibrahim, Thenmozhi Venkatachalam.

**Project administration:** Adel B. Elmoselhi.

**Resources:** Ruqaiyyah Siddiqui, Naveed Ahmed Khan, Rifat A. Hamoudi.

**Software:** Amal Bouzid, Rifat A. Hamoudi.

**Supervision:** Adel B. Elmoselhi, Mohamed Seif Allah.

**Validation:** Mohamed Seif Allah, Ruqaiyyah Siddiqui, Naveed Ahmed Khan.

**Visualization:** Adel B. Elmoselhi, Mohamed Seif Allah.

**Writing – original draft:** Adel B. Elmoselhi, Amal Bouzid, Rifat A. Hamoudi.

**Writing – review & editing:** Adel B. Elmoselhi, Amal Bouzid, Zeinab Ibrahim, Ruqaiyyah Siddiqui, Naveed Ahmed Khan, Rifat A. Hamoudi.

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
