## [Decision Letter · Decision Letter 0]

3 Mar 2023

PONE-D-22-28888Circulating MicroRNAs as Potential Biomarkers of Early Vascular Damage in Vitamin D Deficiency, Obese, and Diabetic PatientsPLOS ONE

Dear Dr. Elmoselhi,

Thank you for submitting your manuscript to PLOS ONE. After careful consideration, we feel that it has merit but does not fully meet PLOS ONE’s publication criteria as it currently stands. Therefore, we invite you to submit a revised version of the manuscript that comprehensively addresses the points of the reviewer.  Please submit your revised manuscript by Apr 17 2023 11:59PM. If you will need more time than this to complete your revisions, please reply to this message or contact the journal office at plosone@plos.org. Please include the following items when submitting your revised manuscript:A rebuttal letter that responds to each point raised by the academic editor and reviewer(s). You should upload this letter as a separate file labeled 'Response to Reviewers'.A marked-up copy of your manuscript that highlights changes made to the original version. You should upload this as a separate file labeled 'Revised Manuscript with Track Changes'.An unmarked version of your revised paper without tracked changes. You should upload this as a separate file labeled 'Manuscript'.

We look forward to receiving your revised manuscript.

Kind regards,

Andre van Wijnen

Academic Editor

PLOS ONE

Journal Requirements:

This study was supported by Sheikh Hamdan Bin Rashid Al Maktoum Award for Medical Sciences, Ref: MRG/38 (ABE received the award)

Reviewers' comments:

Reviewer's Responses to Questions

**Comments to the Author**

1. Is the manuscript technically sound, and do the data support the conclusions?

Reviewer #1: Yes

2. Has the statistical analysis been performed appropriately and rigorously? 

Reviewer #1: Yes

3. Have the authors made all data underlying the findings in their manuscript fully available?

Reviewer #1: Yes

4. Is the manuscript presented in an intelligible fashion and written in standard English?

Reviewer #1: Yes

5. Review Comments to the Author

Reviewer #1: Elmoselhi et al submit an original research article entitled "Circulating MicroRNAs as Potential Biomarkers of Early Vascular Damage in Vitamin D Deficiency, Obese, and Diabetic Patients". They looked at the potential of seric miRNAs as biomarkers of Vitamin D3 deficiency, obesity, and diabetes melitus in 23 middle-aged Emiratis patients. This cohort is very limited. They used a whole transcriptome analysis with qPCR validation for miRNA in plasma samples. They conclude that miR-182-5p; miR-199a-5p; miR-193a-5p; and miR-155-5p were differentially over-expressed in high-risk patients for CVD vs healthy controls.

Comments:

the C1 group (control) is described as having a normal BMI yet they are at a an average of 26 so above the normal level. Can the authors comment?

What controls were performed to check quality of RNAs after extraction?

Can the authors justify the use of miR-19b-3p as housekeeping miRNA ? Can the authors show that this miRNA is the same in all groups?

A recapitulative final figure would be useful

6. PLOS authors have the option to publish the peer review history of their article (what does this mean?). If published, this will include your full peer review and any attached files.

Reviewer #1: No

---

## [Author Response · Author response to Decision Letter 0]

10 Mar 2023

Dear Editor,

I am submitting a revised version of our manuscript, "Circulating MicroRNAs as Potential Biomarkers of Early Vascular Damage in Vitamin D Deficiency, Obese, and Diabetic Patients" with the manuscript number PONE-D-22-28888, for reconsideration by PLOS ONE.

We appreciate the reviewers' valuable comments and feedback, which has significantly improved the quality of our study. We have carefully reviewed the feedback and incorporated the suggestions into our revised manuscript. Our response to reviewer is the attached “Response to Reviewers” file.

We would like to state that although this study was supported by Sheikh Hamdan Bin Rashid Al Maktoum Award for Medical Sciences, Ref: MRG/38 (ABE received the award), the funder had no role in study design, data collection and analysis, decision to publish, or preparation of the manuscript.

Furthermore, we have ensured that the manuscript now meets PLOS ONE's style requirements, including those for file naming. Therefore, in this resubmission, we have included the following files: 1) Response to Reviewers; 2) Revised Manuscript with Track Changes; and 3) Manuscript, which is the revised paper without tracked changes. We believe that the revised manuscript addresses all the reviewers' concerns and meets the high standards of PLOS ONE.

We appreciate the time and effort that the reviewers have invested in providing valuable feedback that has helped us improve our study. We hope that the revised manuscript will be suitable for publication in PLOS ONE.

Thank you for considering our resubmission.

Sincerely,

Adel B. Elmoselhi, MD, PhD

Associate Professor, Basic Medical Sciences Department

College of Medicine, University of Sharjah

P.O. Box 27272 Sharjah – UAE

Phone: +971 6 505 7228 (Office)

Email: amoselhi@sharjah.ac.ae

---

## [Decision Letter · Decision Letter 1]

13 Mar 2023

Circulating MicroRNAs as Potential Biomarkers of Early Vascular Damage in Vitamin D Deficiency, Obese, and Diabetic Patients

PONE-D-22-28888R1

Dear Dr. Elmoselhi,

We’re pleased to inform you that your manuscript has been judged scientifically suitable for publication and will be formally accepted for publication once it meets all outstanding technical requirements.

Kind regards,

Andre van Wijnen

Academic Editor

PLOS ONE

Additional Editor Comments (optional):

Reviewers' comments:

Reviewer's Responses to Questions

**Comments to the Author**

1. If the authors have adequately addressed your comments raised in a previous round of review and you feel that this manuscript is now acceptable for publication, you may indicate that here to bypass the “Comments to the Author” section, enter your conflict of interest statement in the “Confidential to Editor” section, and submit your "Accept" recommendation.

Reviewer #1: All comments have been addressed

2. Is the manuscript technically sound, and do the data support the conclusions?

Reviewer #1: Yes

3. Has the statistical analysis been performed appropriately and rigorously? 

Reviewer #1: Yes

4. Have the authors made all data underlying the findings in their manuscript fully available?

Reviewer #1: Yes

5. Is the manuscript presented in an intelligible fashion and written in standard English?

Reviewer #1: Yes

6. Review Comments to the Author

Reviewer #1: changes are satisfactory changes are satisfactory changes are satisfactory. This is a submision of revised manuscript

7. PLOS authors have the option to publish the peer review history of their article (what does this mean?). If published, this will include your full peer review and any attached files.

Reviewer #1: **Yes: **Laurent METZINGER

---

## [Editor Report · Acceptance letter]

15 Mar 2023

PONE-D-22-28888R1 

Circulating microRNAs as potential biomarkers of early vascular damage in vitamin D deficiency, obese, and diabetic patients 

Dear Dr. Elmoselhi:

I'm pleased to inform you that your manuscript has been deemed suitable for publication in PLOS ONE. Congratulations! Your manuscript is now with our production department. 

Kind regards, 

on behalf of

Dr. Andre van Wijnen 

Academic Editor

PLOS ONE